# Prospective Evaluation of Ultrasound in a Novel Position with MRI Virtual Navigation for MRI-Detected Only Breast Lesions: A Pilot Study of a More Efficient and Economical Method

**DOI:** 10.3390/diagnostics13010029

**Published:** 2022-12-22

**Authors:** Ruixiang Qi, Jianhua Fang, Luoxi Zhu, Yanna Shan, Wei Wang, Chenke Xu, Lingyun Bao

**Affiliations:** 1Department of Ultrasound, Affiliated Hangzhou First People’s Hospital, Zhejiang University School of Medicine, Hangzhou 310006, China; 2Department of Radiology, Affiliated Hangzhou First People’s Hospital, Zhejiang University School of Medicine, Hangzhou 310006, China

**Keywords:** breast ultrasonography, second-look ultrasound, virtual navigation, magnetic resonance imaging, image-guided biopsy

## Abstract

The aim of this study was to evaluate the clinical utility of ultrasound (US) with magnetic resonance imaging (MRI) virtual navigation in a novel prone position for MRI-detected incidental breast lesions. Between June 2016 and June 2020, 30 consecutive patients with 33 additional Breast Imaging Reporting and Data System (BI-RADS) category 4 or 5 lesions that were detected on MRI but occult on second-look US were enrolled in the study. All suspicious lesions were located in real-time US using MRI virtual navigation in the prone position and then followed by US-guided biopsy or surgical excision. Pathological results were taken as the standard of reference. The detection rate of US with MRI virtual navigation was calculated. The MRI features and pathological types of these lesions were analyzed. A total of 31 lesions were successfully located with real-time US with MRI virtual navigation and then US-guided biopsy or localization, and the detection rate was 93.9% (31/33). Twenty-seven (87.1%, 27/31) proved to be benign lesions and four (12.9%, 4/31) were malignant lesions at pathology. Of the 33 MRI-detected lesions, 31 (93.9%, 31/33) were non-mass enhancements and two (6.1%, 2/33) were masses. This study showed that real-time US with prone MRI virtual navigation is a novel efficient and economical method to improve the detection and US-guided biopsy rate of breast lesions that are detected solely on MRI.

## 1. Introduction

Breast cancer is one of the most common cancers and the leading cause of cancer deaths in females, with an incidence of 24.2% and mortality of 15.0% as reported in Global Cancer Statistics 2018 [1]. The early diagnosis of breast cancer is the key to reducing mortality. Ultrasound (US) is currently the preferred method for early breast cancer screening in China. Although the detection rate of US for the diagnosis of early breast cancer is increasing with the development of technology, some atypical or occult lesions are easily misdiagnosed or the diagnosis is missed.

Contrast-enhanced magnetic resonance imaging (CE-MRI) has progressively proved to have a substantial role in the detection and evaluation of breast cancer owing to its high sensitivity. When a suspicious lesion is detectable on a breast MRI but not on a routine US, an MRI-guided biopsy of the lesion is usually recommended. However, MRI-guided breast percutaneous sampling has many inadequacies. The procedure has a high cost and is time-consuming, and it requires the injection of contrast material. Moreover, it may be difficult to access lesions located posteriorly or deeply [2].

In view of the limitations of MRI-guided biopsy, second-look US is recommended as the preferred next step. Second-look US is a targeted breast US to identify ultrasonic correlated lesions detected on breast MRI. Various studies have revealed that the detection rate of suspicious lesions with second-look US is diverse, ranging from 22.6% to 82.1%; the reason for such a wide range may be due to differences in experience between operators or inconsistent equipment [3]. 

A new technique has been developed recently to solve these problems by coordinating the sonographic and MRI multiplanar reconstruction (MRI-MPR) images at the same section in real-time [4]. In addition, a US-guided biopsy can then be performed to sample lesions occult on second-look US to make a definitive diagnosis. In view of negative findings on the initial US, the lesion on second-look US is still invisible when real-time US with US/MRI fusion virtual navigation is performed. 

Several studies have reported that virtual navigation in the supine position is an effective method that increases the sonographic detection rate and biopsy rate of occult lesions detected by MRI [5,6,7,8]. However, an additional MRI examination in the supine position is needed and it is costly and time-consuming, requires the injection of contrast agent once again, and may be unavailable and uncomfortable [9]. Moreover, some lesions do not show on MRI during supine position examination, and changes in the examination position may cause the breast to deform, resulting in differences in the location of the lesions. 

To determine a more efficient and economical method to evaluate the lesions, which were MRI-detected only but occult on conventional and second-look US, we used fusion image technology combining US/MRI in real-time in a novel position with the MRI data already obtained, without an additional MRI examination. The purpose of this study was to explore the clinical value of real-time US with MRI virtual navigation in a novel position and to analyze the imaging features and the pathological results of these MRI-detected incidental breast lesions.

## 2. Materials and Methods

### 2.1. Study Population

Between June 2016 and June 2020, a total of 30 patients (mean age ± standard deviation, 44 ± 7 years; range 33–65) from our hospital were prospectively evaluated. This study was approved by our hospital’s institutional review board (IRB approval number: 2018-HZSY-XJS-036-01). Written informed consent was obtained from all patients. A flowchart of the study is shown in Figure 1.

The enrolled patients went to the hospital for US examination due to chest pain, nipple discharge, or nodules found in physical examination or family history of breast cancer, but the initial US evaluation was negative and then MRI examination was planned. MRI examination was performed in the following cases: (1) no abnormal findings on initial US, or suspected or insufficiently evaluated lesions (Breast Imaging Reporting and Data System (BI-RADS) category 0) on X-ray; (2) ultrasonography revealed category 4 or more lesions in the contralateral breast; and (3) past history of atypical hyperplasia or family history of breast cancer.

The study inclusion criteria were that lesions detected by MRI and assessed as BI-RADS category 4 or 5 by two radiologists (with more than 5 years experience in breast imaging), but without corresponding lesions detected by conventional US, and which were still not evident on second-look US. The exclusion criteria were no pathological results or lack of follow-up.

### 2.2. Sonographic and MRI Examinations

All MRI examinations were performed in the second week of the menstrual cycle in premenopausal women, in the routine prone position with the administration of a contrast medium as previously described [10]. All patients underwent clinical palpation and conventional imaging examination before the MRI examination. When an additional lesion was detected on MRI examination, a second-look US was performed. 

Second-look US was performed using a high-frequency linear probe 18L6HD (Acuson S2000 Ultrasound System, Siemens, Germany) while the patient was in the supine position and the bilateral breasts and armpits were fully exposed. For lesions not found during the initial US scan, a second-look US examination was performed by a sonologist with more than 10 years of experience in ultrasonographic diagnosis of breast diseases.

### 2.3. Real-Time US with Virtual Navigation

The additional lesion detected on breast MRI was re-evaluated after a multidisciplinary discussion (including a breast surgeon, radiologist, sonologist, and pathologist), and reconfirmed as BI-RADS 4 or above lesions before real-time US with virtual navigation. All patients accepted US-guided biopsy or localized excision.

Real-time US fusion with the pre-acquired MRI images was performed using a commercially available US system (MyLab Twice, Esaote, Italy) equipped with the Virtual Navigator module. The device consists of two electromagnetic sensors adhered to the US probe (LA 523, Operating Bandwidth: 4–13 MHz). When the device was set up, the MRI digital imaging and communications in medicine data were transferred to the US system, after which the MRI and US images were simultaneously displayed on a monitor, and the two images registered in the same position and direction [9].

During the synchronization process, US images and MRI-MPR were alternately displayed at the same time as the operator was moving the probe on the breast. Generally, the nipple on the side being examined was easily recognized by both US and MRI and it was often recommended as a reference point for co-registration. Moreover, the spatial relationship between the known lesion and surrounding vessels can also be used as an additional reference to enable a more precise matching. In addition, internal anatomical landmarks such as blood vessels, fat lobules, and Cooper’s ligaments can be used for co-registration between the two modalities. When either image is misaligned, it requires resynchronization. 

In the present study, real-time US with virtual navigation processes were performed in the prone position using a special prone positioning frame (Figure 2). The patient lay prone to the device, allowing both breasts to droop naturally in the hole above, and the operator performed the US examination below the hole. After axial alignment was performed at the nipple level, and the contour of the breast was aligned, the surrounding tissues such as mammary fascia, blood vessels, or fat lobules were then matched. The Virtual Navigator system calculated spatial information and displayed an MRI-MPR image corresponding to the US image in real time. Subsequently, the operator moved the transducer gradually to narrow the scan range on US images and search the location of the suspected corresponding lesion (Figure 3). When a suspicious lesion was identified with real-time US with MRI virtual navigation, we analyzed its sonographic features and evaluated the lesion’s location and surrounding tissues and marked it. Then, the patient was put in the supine position, and a US-guided core needle biopsy or needle localization was performed referring to the area identified on the prone US.

## 3. Results

A total of 30 patients with 33 lesions were included in the study. Only 1 (3.3%) patient had composition b (scattered fibroglandular) according to the American College of Radiology BI-RADS edition 2013 breast composition categories, 17 (56.7%) patients had composition c (heterogeneously dense) and 12 (40.0%) patients had composition d (extremely dense).

All 33 lesions were detected on breast MRI only and were invisible on second-look US. All suspicious lesions were re-evaluated as BI-RADS category 4 or more by multidisciplinary discussion. The characteristics of all additional lesions detected by MRI are summarized in Table 1. Of 33 additional MRI-detected lesions, 2 (6.1%) were masses, and 31 (93.9%) were non-mass enhancements (NME). 

Two lesions displayed no indications on US with virtual navigation and the location of the breast lesion was too deep and there were no anatomical markers or abnormalities around it, and US-guided biopsy of these lesions was not performed. Follow-up MRI examinations were performed on these two lesions and there was no change during more than 2 years of follow-up. Real-time US with virtual navigation detected 93.9% (31/33) of breast lesions detected by MRI alone. 

Pathologic confirmation was eventually obtained for 31 lesions. A total of 22 of the lesions were surgically resected, and post-procedural MRI showed that the lesions disappeared. Nine lesions had US-guided biopsy performed only, and the pathological findings were benign, with no change in the lesions on follow-up MRI for two years. The pathological types of these 31 lesions detected by real-time US with MRI virtual navigation are shown in Table 2. Among them, 87.1% (27 cases) were benign lesions (18 were adenosis, 5 intraductal papilloma, 2 fibroadenoma, and 2 atypical hyperplasia), and 12.9% (four cases) were malignant lesions. The pathological type of all malignant lesions was ductal carcinoma in situ (DCIS). Two of the malignant lesions were multifocal lesions in the unilateral breast of the same patient. The pathology showed that the maximum diameter was 0.6 cm and 0.5 cm, respectively. One malignant lesion was a widespread hypoechoic area, which was pathologically demonstrated as DCIS with a maximum diameter of 3.0 cm. Another malignant lesion was a wide-ranging lesion of the bilateral breast in the same patient and the lesion in the left breast was located with real-time US with MRI virtual navigation (Figure 4). The pathological type was DCIS with a maximum diameter of 1.4 cm in the left breast and 2.0 cm in the right breast. All malignant lesions underwent radical resection, and the final histopathological diagnosis was the same as before. These four malignant lesions were all non-mass enhancements on MRI. 

## 4. Discussion

As reported previously, MRI has high sensitivity in detecting breast lesions, but the specificity is only moderate, with pooled weighted estimates of sensitivity and specificity of 0.90 and 0.72, respectively [11]. When additional lesions are detected by breast MRI which are occult on conventional US, second-look US is usually recommended to evaluate the suspicious lesions. A large meta-analysis of 2201 lesions reported that second-look US proved to be valuable for further evaluation of suspicious lesions on breast MRI images, with a 57.5% pooled detection rate (range, 23–82% of cases) [3]. 

On the other hand, second-look US is strongly operator dependent, and a considerable number of lesions detected on MRI are not visible on second-look US, as the reported rate ranges from 13% to 54% [12]. Consequently, an MRI-guided biopsy is performed to make a definitive diagnosis of the lesions invisible on second-look US. Nevertheless, MRI-guided breast biopsy has several limitations: it is technically difficult, it cannot perform real-time visualized guidance, the rate of false positives is high, it is expensive and time-consuming, and it requires the injection of contrast agent [12]. 

Recently, virtual navigation has been developed, and several studies have shown that the use of supine breast MRI images for co-registration during real-time US examination has good clinical application [13,14,15,16]. Research by Uematsu et al. showed that of 78 MRI-detected lesions included in the study, 50 (64%) lesions were detected with second-look US; 24 (31%) were found with real-time US with virtual navigation in the supine position, and the remaining 4 (5%) lesions were not identified on the second breast MRI. In addition, of 24 lesions found with US with virtual navigation, 17 (71%) were benign and 7 (29%) were malignant [13]. US/MRI fusion virtual navigation was used to detect the corresponding lesions detected on MRI only, and then US-guided biopsy was performed possibly to sample lesions that were occult on the US to make a definitive diagnosis. US-guided biopsy has more advantages than MRI-guided biopsy. It is convenient to operate, highly feasible, and economical, with real-time visualization of lesions and great patient comfort [2]. 

Of note, in the present study, an important innovation was that we used conventional prone breast MRI data, and real-time US with virtual navigation was performed in the prone position without additional MRI examination. In previous studies, to ensure that the breast position was consistent with the US examination, the patient underwent a special MRI examination in the supine position. However, the breast coil used in the MRI examination in the supine position was not optimized. Its voxel size was large, and the spatial resolution was low, and sometimes the suspicious lesions found in the prone position MRI could not be detected on supine MRI. The position of breast lesions and the nipple will vary greatly depending on the patient’s position. Kang et al. [12] reported that 13% of lesions detected on conventional prone MRI were not observed on supine MRI. Moreover, a special supine MRI examination is time-consuming and costly and requires another injection of contrast agent [9]. Compared with supine virtual navigation, our method can avoid an additional MRI examination in the supine position and save time and reduce costs.

Conventional breast MRI imaging data are obtained in the prone position, while conventional US and second-look US are performed in the supine position. The difference between the supine and prone positions can lead to breast deformation, resulting in a discrepancy in lesion location. A previous study [17] reported that the median lesion dislocation of breast MRI lesions in the prone position was roughly 3–6 cm along three orthogonal directions in comparison with supine MRI. In our study, real-time US with virtual navigation was innovatively performed in the prone position to obtain the same position as the regular MRI examination. Thus, we achieved the best match of anatomical position while fusing US and MRI images. Prone breast MRI is recommended as a routine diagnostic examination in view of the image quality in the supine position which is not as good as that in the prone position due to respiratory motion or heartbeat artifacts and the use of inappropriate coils [6].

Our study found that the detection rate of real-time US with MRI virtual navigation in the prone position was 93.9%, which was a little higher than conventional supine virtual navigation [18,19]. Research by Nakano et al. [18] revealed that of 67 MRI-detected additional lesions, 90% of these lesions were identified using real-time US with virtual navigation in the supine position, whereas only 30% were detected by second-look US. Pons et al. [19] found that the detection rate for incidentally MRI-detected lesions on real-time supine-MRI navigated US was 90.7%, which was significantly higher than conventional US (43%). Real-time US with MRI virtual navigation in the prone position can also significantly increase the detection rate of negative lesions on conventional US and can be an effective method for the evaluation of such lesions. Compared with the supine position, navigation in the prone position may be more comfortable, economical, rapidly, and efficient.

In the present study, 93.9% of the 33 lesions detected on MRI but negative on second-look US were NME lesions. An NME lesion detected on MRI usually presents as a localized hypoechoic area or fuzzy area with a structural disorder or catheter abnormality on US [20]. It is difficult to confirm the diagnosis with US because there is no obvious tumor entity or mass effect. Virtual navigation can synchronize fusion US and MRI images and can display the suspicious lesions detected on MRI in real-time US, which can then be positioned or sampled under US guidance (Figure 5). Studies by Uematsu and Watanabe et al. [8,13] have shown that virtual navigation can improve the US detection rate and biopsy rate of breast lesions found on MRI only.

In this study of 33 additional MRI-detected lesions which underwent real-time US with virtual navigation, 2 failed and the remaining 31 lesions were detected with US with virtual navigation, and US-guided biopsy or surgery was performed. A total of 22 of the lesions were surgically resected, and post-procedural MRI showed that the lesions disappeared. Nine lesions were performed US-guided biopsy only, and there was no change during the follow-up MRI. Which verify that the US-detected area via MRI virtual navigation was actually consistent with the enhancing area on MRI. According to the pathological results, 4 (12.9%) were malignant and 27 (87.1%) were benign with 2 high-risk lesions. This shows that abnormally enhanced MRI lesions that were invisible on US tend to be benign lesions. Real-time US using US/MRI fusion virtual navigation can reduce unnecessary surgery and trauma caused by over-diagnosis due to breast MRI. As the detection rate of conventional US in our center is high, only a few cases of invisible sonographic lesions are included in this study. This may be one reason for the low malignancy rate in our study.

There are also several limitations in this study. Breast tissues are soft and easily deformed so there will be spatial displacement causing discrepancies and misalignment during co-registering. Accurate alignment is difficult between two different imaging modalities [21]. Nevertheless, the operator can use some skills to reduce these variants, such as avoiding excessive pressure on the breast during the US examination, keeping the probe perpendicular to the skin surface, and accurately aligning the body marker on MRI images [5]. This requires the operator to have a wealth of image interpretation capabilities and a long learning curve. Another disadvantage is that the technique is not suitable for patients with breast hypertrophy. It is difficult to locate deep lesions in large breasts, and the prone position can increase the inspection depth, resulting in difficulty in distinguishing far-field images. In this situation, low-frequency probes may be useful. Finally, we need to know that this technique can narrow the scope of the lesion location rather than completely accurate positioning.

## 5. Conclusions

This study shows that real-time US with prone MRI virtual navigation is a novel efficient and economical method to improve the detection and US-guided biopsy rate of breast lesions that are detected solely on MRI. Real-time US with prone MRI virtual navigation is an accurate and feasible method to complement second-look US and may be used as the “third-look US” for breast lesions.

## Figures and Tables

**Figure 1 diagnostics-13-00029-f001:**
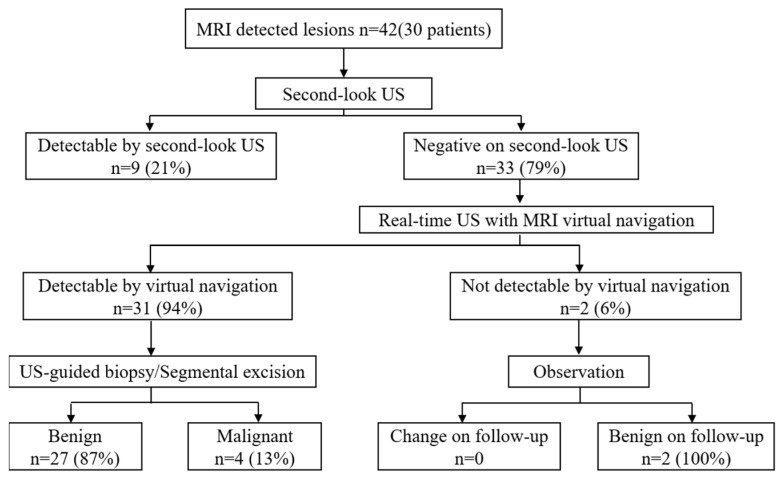
Flowchart of this study.

**Figure 2 diagnostics-13-00029-f002:**
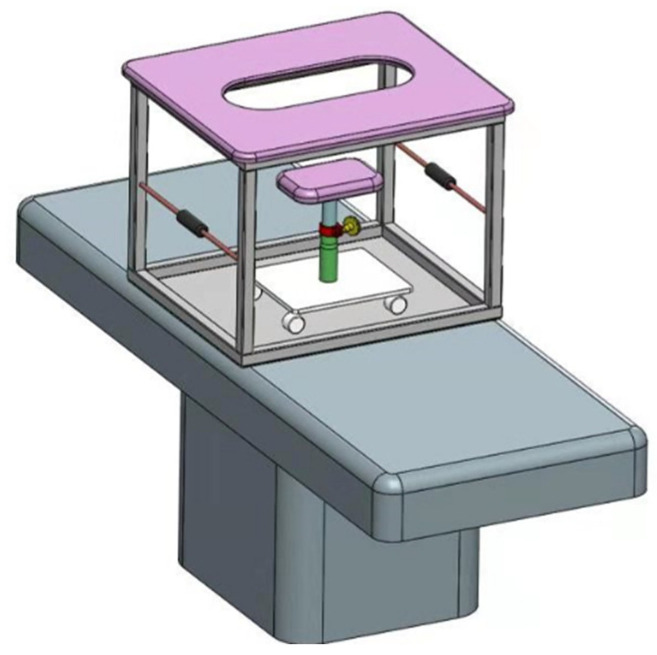
A special prone positioning frame designed specifically for real-time US with prone MRI virtual navigation. US = ultrasound; MRI = magnetic resonance imaging.

**Figure 3 diagnostics-13-00029-f003:**
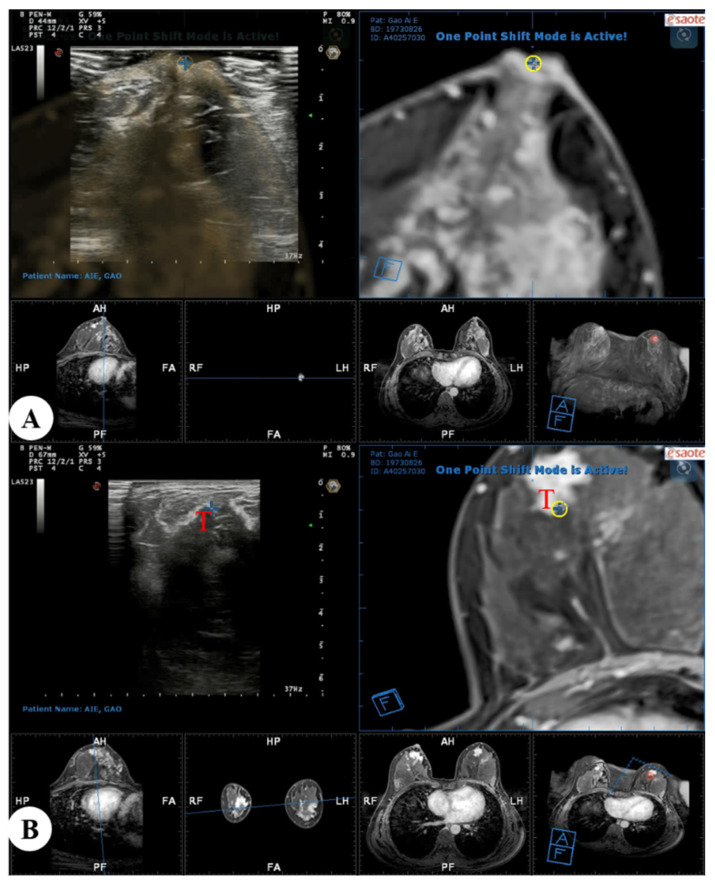
Real-time US with virtual navigation was performed in the prone position: (**A**) A co-registration process between real-time US and MRI-MPR images was performed with the nipple as a reference point, and the contour of the breast and internal anatomic landmarks were aligned; (**B**) Search and localization using real-time US with virtual navigation of the corresponding suspected enhanced lesion (T, target) identified on MRI alone. US = ultrasound; MRI = magnetic resonance imaging; MPR = multiplanar reconstruction.

**Figure 4 diagnostics-13-00029-f004:**
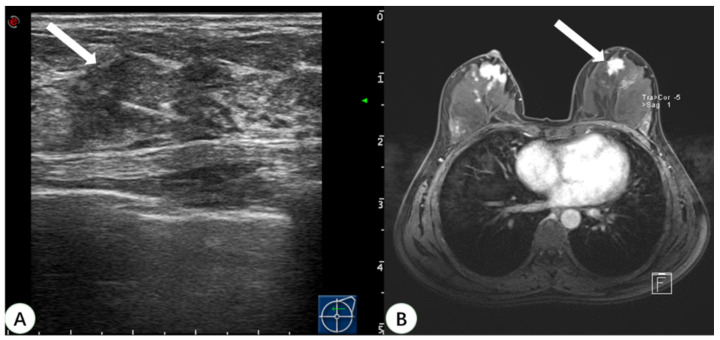
A 46-year-old woman with dense breast tissue underwent virtual navigation: (**A**) Real-time US with virtual navigation detected the corresponding lesion (arrow) in the left breast at the 12-o’clock position 2 cm from the nipple, which underwent ultrasound-guided biopsy; (**B**) CE-MRI showed a suspicious enhanced irregular lesion (arrow) in the upper left breast, undetected at second-look US. Pathology obtained by US-guided biopsy with virtual navigation demonstrated a ductal carcinoma in situ. US = ultrasound; CE-MRI = contrast-enhanced magnetic resonance imaging.

**Figure 5 diagnostics-13-00029-f005:**
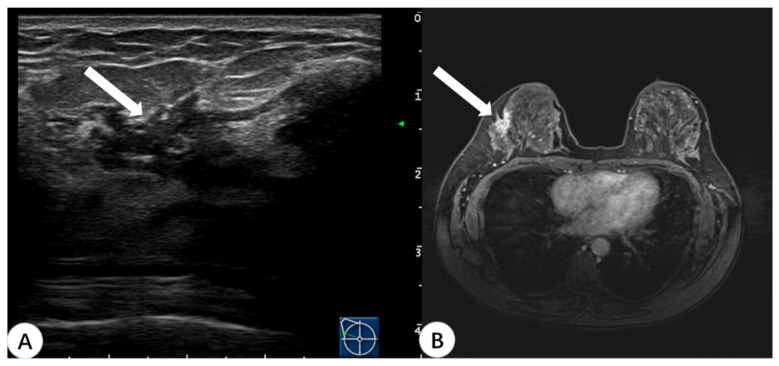
A 44-year-old woman with dense breast tissue underwent virtual navigation: (**A**) Real-time US with virtual navigation revealed the corresponding lesion (arrow) in the right breast at the 10-o’clock position 5 cm from the nipple, which underwent ultrasound-guided biopsy; (**B**) CE-MRI showed a non-mass enhanced lesion (arrow) in the upper outer right breast, undetected on second-look US. Pathology obtained by ultrasound-guided biopsy with virtual navigation demonstrated sclerosing adenopathy with intraductal papilloma formation. US = ultrasound; CE-MRI = contrast-enhanced magnetic resonance imaging.

**Table 1 diagnostics-13-00029-t001:** The MRI characteristics of 33 additional lesions detected with MRI only.

Lesion Characteristic	No. of Lesions	Benign	Malignant
Lesion type			
Mass	2	2	
NME ^1^	31	27	4
Delayed phase enhancement			
Persistent	17	17	
Plateau	14	11	3
Washout	2	1	1

^1^ NME, non-mass enhancements.

**Table 2 diagnostics-13-00029-t002:** Pathological results and MRI follow-up results of 33 included lesions.

Pathology	N	%
Adenopathy	18	54.5
Fibroadenoma	2	6.1
Intraductal papilloma	5	15.1
Atypical hyperplasia	2	6.1
Ductal carcinoma in situ	4	12.1
MRI follow-up without pathological results	2	6.1

## Data Availability

Data available on request from the authors.

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
