# Peer review of "Prospective Evaluation of Ultrasound in a Novel Position with MRI Virtual Navigation for MRI-Detected Only Breast Lesions: A Pilot Study of a More Efficient and Economical Method"

_diagnostics, 2022, doi:10.3390/diagnostics13010029_

Round 1
Reviewer 1 Report
Authors described real-time US with MRI virtual navigation would be economic method. So, potential economic advantage of this method should be shown in results or described in discussion.
Reviewer 2 Report
This technology approach described in this paper is very interesting and relevant. If this study can be performed on a larger scale and involve more countries outside of China, then this technology would be very popular and can be practice changing.
Is it possible for you to have more images of the machine and patient's positioning diagram to allow better appreciation of the two exams. Is this a software that can be downloaded on any computer? Or is it a workstation in addition to the US machine?
How labor intensive is this for the sonographer or radiologist who is scanning the breast?
I would highly recommend that the study be performed with larger sample size. Nice work!
